# Using Single-Case Experimental Design and Patient-Reported Outcome Measures to Evaluate the Treatment of Cancer-Related Cognitive Impairment in Clinical Practice

**DOI:** 10.3390/cancers15184643

**Published:** 2023-09-20

**Authors:** Robert J. Ferguson, Lauren Terhorst, Benjamin Gibbons, Donna M. Posluszny, Hsuan Chang, Dana H. Bovbjerg, Brenna C. McDonald

**Affiliations:** 1Division of Hematology/Oncology, Department of Medicine, University of Pittsburgh, Pittsburgh, PA 15232, USA; poslusznydm@upmc.edu (D.M.P.); shenchang116@gmail.com (H.C.); 2Department of Occupational Therapy, School of Health Rehabilitation Sciences, University of Pittsburgh, Pittsburgh, PA 15260, USA; lat15@pitt.edu; 3Department of Family Medicine, School of Medicine, University of Pittsburgh, Pittsburgh, PA 15260, USA; bfg10@pitt.edu; 4UPMC Hillman Cancer Center, Department of Psychiatry, Biobehavioral Cancer Control Program, University of Pittsburgh, Pittsburgh, PA 15232, USA; bovbjergdh@upmc.edu; 5Department of Radiology and Imaging Sciences, Indiana University School of Medicine, Indianapolis, IN 46202, USA; mcdonalb@iupui.edu

**Keywords:** cancer survivorship, cognition, cognitive-behavioral therapy, single-case experimental design

## Abstract

**Simple Summary:**

After cancer diagnosis and treatment, many patients report difficulty with cognitive functions such as learning, memory, and attention. Cancer-related cognitive impairment (also known as CRCI) can lead to problems in work, social life, and other daily activities. Research on the treatment of CRCI is ongoing, and one approach, cognitive-behavioral therapy or CBT, may be helpful. This article describes how to track symptoms with patient-reported outcome measures for individual cancer survivors who engage in CBT or other CRCI treatments in clinical practice. This system may help determine if treatments are effective and improve real-world patient outcomes.

**Abstract:**

Cancer-related cognitive impairment (CRCI) affects a large proportion of cancer survivors and has significant negative effects on survivor function and quality of life (QOL). Treatments for CRCI are being developed and evaluated. Memory and attention adaptation training (MAAT) is a cognitive-behavioral therapy (CBT) demonstrated to improve CRCI symptoms and QOL in previous research. The aim of this article is to describe a single-case experimental design (SCED) approach to evaluate interventions for CRCI in clinical practice with patient-reported outcome measures (PROs). We illustrate the use of contemporary SCED methods as a means of evaluating MAAT, or any CRCI treatment, once clinically deployed. With the anticipated growth of cancer survivorship and concurrent growth in the number of survivors with CRCI, the treatment implementation and evaluation methods described here can be one way to assess and continually improve CRCI rehabilitative services.

## 1. Introduction

Cancer-related cognitive impairment (CRCI) is a prevalent, chronic problem of mild to moderate cognitive change resulting from disease and/or treatment, negatively affecting survivor occupational and social role function [1,2,3,4,5,6,7,8]. While the exact prevalence of CRCI is not known, estimates are that nearly half of all survivors may have long-term mild/moderate cognitive impairment for years after the completion of cancer treatment [1,9]. With 18 million survivors in the US [10] and 19.3 million newly diagnosed individuals worldwide [11] at the time writing, the problem of CRCI is broadening. The number of cancer survivors is projected to increase substantially, given the combination of an aging population, improvements in early detection, and the advancement of new cancer therapies [7,12]. In light of these factors, identifying and delivering effective CRCI treatment is a priority [13].

Causes of CRCI are thought to be multiple [1,14,15], with numerous mechanisms explored, including the inhibition of hippocampal neurogenesis, microvascular damage, oxidative stress, direct neurotoxicities of chemotherapy agents, hormonal disruption with hormonal therapies, and inflammation [1]. Consistent with this causal diversity, various interventions have been developed and evaluated. These include pharmacologic approaches, exercise, repetitive cognitive practice with computerized programs, and cognitive behavioral therapy (CBT) [16,17,18,19,20,21].

While research on CRCI treatment continues, some treatments are beginning to be used in clinical practice [22]. Understanding how effective any treatment of CRCI may be under real-world clinical conditions in contrast to the constraints of selective research eligibility criteria used in randomized clinical trials (RCTs) is critical for (1) optimizing patient clinical outcomes and (2) estimating the treatment’s value for patients and healthcare professionals [23,24]. The aim of this article is to propose a replicable method of using single-case experimental design (SCED) and patient-reported outcome measures (PROs) in clinical settings. Data generated by SCED methods can help refine CRCI treatments through the identification of the most effective treatment components matched to survivor characteristics that yield the best outcomes [25,26]. This informs CRCI care in clinical oncology [24] and helps address unmet survivorship needs [27]. In this article, we illustrate the use of this system with current cognitive-behavioral therapy (CBT) for CRCI (memory and attention adaptation training; MAAT) [28,29]. We also present an MAAT case illustration using SCED with freely available PROs and discuss the benefits and limitations of this method.

### 1.1. PROs and Single-Case Experimental Design

#### 1.1.1. PROs

PROs have been proposed to be used in routine oncology practice for over a decade with the dual purpose of clinical monitoring of patient distress (e.g., pain, treatment toxicity, and psychological distress) and as a metric of health outcomes [30]. Recent guidelines propose PRO monitoring in three cancer care stages. These include (1) early detection and management of treatment toxicities, (2) early detection and management of cancer recurrence and complications, and (3) tailoring supportive cancer care [31]. The focus of PRO monitoring in this article is to evaluate clinical outcomes of CRCI management in supportive cancer care settings (e.g., behavioral oncology). For a more extensive overview of PRO use in cancer care, see Di Maio et al. [31] and the University of North Carolina’s Patient-Reported Outcomes Core: https://pro.unc.edu (accessed on 9 September 2023).

For PRO monitoring of CRCI, we recommend PROs from the Patient-Reported Outcomes Measurement Information System^®^ (PROMIS). PROMIS measures are public domain and freely available through www.healthmeasures.net (accessed on 19 May 2023). However, observable behaviors, such as the frequency of misplacing personal items, such as keys or mobile phones, reported by the patient or significant other, can also be used as an outcome measure (or a dependent variable in SCED; see below) [32]. The eight-item PROMIS v2.0, Cognitive Function-Short Form 8a (CF8), is a widely used and valuable PRO of CRCI [33]. It is brief, minimizes the questionnaire burden, and can be completed weekly by survivors to track their treatment progress (it asks respondents to report on the “last 7 days”). The CF8 items are derived from the large PROMIS cognitive item pool and are identical to some items from the Functional Assessment of Cancer Therapy–Cognitive (FACT–Cog) Perceived Cognitive Impairments scale (PCI) [34], and the measure is internally consistent with good construct validity [33,35]. CF8 item ratings are scaled from 5 (never) to 1 (very often, several times a day), with a possible raw score range of 8 to 40. A T-score conversion table was published in the PROMIS cognitive function manual (mean = 50; sd = 10), in addition to an online HealthMeasures Scoring Service (https://www.assessmentcenter.net/ac_scoringservice (accessed on 19 May 2023)). As with all PROMIS measures, the higher the score, the more the construct is measured (higher = better cognitive function). T-scores allow for easy interpretation and standardized comparisons among other PROMIS measures, such as fatigue, depression, or anxiety, all of which may covary with or influence CF8 scores. Last, the PROMIS cognitive function manual recommends a ½ standard deviation (or five T-score points) change as a minimal clinically important difference (MCID) [36,37]. That is, if a survivor meets or exceeds a five T-score point range in improvement from baseline, the change is considered clinically significant. A broader discussion on PROMIS MCIDs can be found at: http://www.healthmeasures.net/score-and-interpret/interpret-scores/meaningful-change (accessed on 19 May 2023) and in relevant reviews [36,37,38].

#### 1.1.2. SCED for Clinical Practice: Rationale

SCED methods, when combined with routine oncology PRO monitoring in clinical settings, offer ways to empirically evaluate treatments for CRCI, such as MAAT. This can be carried out with individual cases or groups of cases [32,39]. We propose taking advantage of existing PRO monitoring systems and using SCED and contemporary SCED statistical methods [32,39,40,41,42,43]. This allows for continual data analysis of CRCI treatments in order to modify and improve them or to better determine which cancer survivors derive benefit and those who do not. While the gold standard for establishing the efficacy of an intervention is the RCT, SCED offers a means of evaluating CRCI treatment once it is deployed in clinical practice.

There are six points that support the rationale for SCED analysis of clinical outcomes:(1)SCEDs allow for evaluation of all survivors referred for CRCI treatment with any cancer diagnosis with one or more comorbidities—this provides important information about survivors who might not match those carefully selected in RCTs for MAAT or other CRCI treatments [42,44,45,46];(2)SCEDs allow for the evaluation of intra-subject variability with multiple assessments before, during, and after the course of treatment, whereas large RCTs typically evaluate less frequently (e.g., baseline, post-treatment, and follow-up). SCEDs thus permit a clearer understanding of the trajectory and timing of treatment response and what factors contribute to response or non-response [39,40,42];(3)SCEDs can be used to continually inform and improve treatment and thus keep up with rapid technological advances of newly developed cancer therapies that may induce CRCI [22,25,26,47,48];(4)SCEDs can provide important clinical information about outcomes that can contribute to the knowledge base of clinical science, *especially in rare tumor types,* where large RCTs are not easily conducted (e.g., gastrointestinal stromal tumors) [49];(5)SCED data take less time to be gathered than RCTs, which typically have delays in start-up, completion, and results reporting [50], and SCEDs are less expensive than RCTs.(6)SCED evaluation can be within a single patient or aggregated across patient cohorts with multiple SCED reports in one or more forms of cancer treatment. For example, examining outcomes of several SCED reports of individuals with CRCI who have undergone chimeric antigen receptor therapy (CAR T-cell) [22]. SCED may also help identify which survivors benefit most, least, or not all from MAAT [26,40] and explore possible related effects of other survivor problems, such as medical or psychological comorbidities [26,32].

#### 1.1.3. SCED Methods and Brief Overview

Below, we review two basic SCEDs that can be used in clinical settings. We use MAAT as an example of CRCI treatment and CF8 as the target outcome. A case illustration is presented later. For more detailed information on SCED, see the excellent classic work of Barlow and Herson [32,39] and a more contemporary review of SCED selection by Manolov [42].

The first SCED, the A-B or “bi-phasic” design (Figure 1) [39], is often used in clinical settings due to its simplicity and practical application. “Phase A” refers to the pre-treatment baseline with at least three baseline data points, and “Phase B” refers to the treatment phase, with multiple data points across the course of MAAT or other intervention. The reason for at least three baseline data points is to identify any possible variation in the target symptom with a trend of improvement prior to the initiation of treatment. This would lead to a question of whether treatment was a primary influence of symptom improvement versus another factor, such as recovery due to treatment expectation or placebo. Follow-up data assessments can also be conducted well after treatment at defined time points, such as monthly follow-up assessments, to evaluate outcome durability in CF8 scores. That is, does the patient’s outcome maintain the MCID well after treatment ends? The A-B design is used instead of an “A-B-A” design (baseline, treatment, then withdrawal of treatment, such as drug treatment) because it is not possible to “withdraw” effects of CBT treatment once an individual has modified thinking or acquired and applied new knowledge and compensatory behaviors. It is, of course, possible and likely that some new behaviors or cognitive skills fade after active treatment or regular interaction with a therapist ends. Thus, follow-up assessments are good practice for assessing treatment durability. In summary, A-B SCED seems the most practical and least intrusive model to incorporate in clinical practice.

The second design is a derivative of A-B SCED. The multiple-baseline (MBL) across-subject design can be conducted when a small group of three or four survivors start CRCI treatment at staggered time points but simultaneously begin baseline CF8 administration (Figure 2). Each survivor has successively longer baselines to control for history and pre-treatment variation in cognitive symptoms, thus allowing a more rigorous interpretation of baseline to treatment phase change [39,51]. To add scientific rigor, if survivors are willing and consent, randomizing individuals to an assigned baseline time period can enhance confidence in results to account for treatment expectancy factors or eagerness-based “flight into health” on patient-reported outcomes.

The MBL across-subject method can be used to evaluate survivors in different categories of interest, such as individuals with rare tumor types for whom large RCTs would be impractical, such as gastrointestinal stromal tumors [49]. Similarly, this approach can be useful for small groups of individuals who receive the same new cancer treatment [22] (e.g., survivors completing a new anti-PD1/PDL1 therapy).

Another MBL method is multiple-baseline assessment across symptoms (or behaviors) in one survivor. For example, PROMIS measures of fatigue, sleep disturbance, and cognitive function can be concurrently measured during the same baseline time span. All three outcomes are plotted on three graphs for the same survivor for comparison (Figure 3). Since PROMIS measures in different domains have T-score conversions, direct quantitative comparisons can be made. MBL across symptoms can help identify the covariation of symptoms related to CRCI and the relative impact on treatment. For example, it could be that fatigue covaries strongly with the improvement in CF8 scores for some individuals participating in MAAT but not others (note higher CF8 scores = better cognitive function but lower fatigue/sleep disturbance = less fatigue/sleep disturbance).

Any practitioner or group of clinicians treating cancer survivors for CRCI can use PRO monitoring and SCEDs to evaluate CRCI treatment effectiveness, identify which cancer survivors are most responsive to treatment, and potentially identify targets for treatment improvement. In the Materials and Methods Section, we describe MAAT in greater detail and electronic data capture methods using REDCap that can be used in practice. We also provide instructions on how to set up a REDCap program that deploys weekly PROs in the Appendix A. MAAT is used for purposes of illustration in this article to demonstrate the “fit” of SCED design to the CRCI treatment in terms of its number of visits and desired data points collected for the most valid results possible. We also describe the methods of analyzing, interpreting, and reporting SCED results. A case illustration is presented in the subsequent Results section to demonstrate the utility of SCED and outcome analysis in a single cancer survivor.

## 2. Materials and Methods

### 2.1. MAAT

MAAT is not the only treatment of CRCI under development but has been constructed and evaluated over the last 2 decades. MAAT has been shown to improve patient-reported and objective neurocognitive outcomes and QOL in RCTs with breast cancer survivors [52,53,54] and has been shown to improve neurocognitive outcomes in trials of men and women after traumatic brain injury and those with epilepsy. These results suggest generalizability across diverse patient populations with differing underlying etiology for cognitive problems [55,56]. A large multi-site, active control randomized trial is currently underway with breast cancer survivors (NCT 04586530), and MAAT has been modified for older adults with an active control RCT in progress (NCT 04669301).

MAAT is a brief CBT and consists of 8 weekly visits of 45–53 min each, with a standardized clinician manual [28] and survivor workbook [29]. It is designed for telehealth delivery to minimize the travel burden for survivors [54,57]. On a practical level, MAAT takes a “compensatory strategy” approach that teaches new skills to aid performance on daily tasks for which memory is critical [58]. It emphasizes the acquisition of adaptive behavioral, emotional regulation, and cognitive skills to optimize cognitive performance and emotional coping with cognitive dysfunction brought on by the biological impacts of cancer and cancer therapy [28,58,59]. The compensatory strategy approach has been found to generalize (or “transfer”) across multiple settings better than repetitive practice (computerized) interventions that limit improvement to trained tasks [59,60,61,62,63,64]. A deeper theoretical understanding of MAAT is based on a diathesis–stress model and social learning theory. This model considers complex biological etiology (e.g., chemotherapy exposure) and multiple biopsychosocial factors that perpetuate the maintenance of CRCI cognitive symptoms and distress. Memory and attention are affected by multiple interacting neurocognitive systems. Affective states, physiological arousal, sensory acuity (i.e., hearing and seeing), and environment all influence orientation, attention, encoding, speed of processing, retrieval, recognition, and recall. Under routine, low-demand conditions, these interacting neurocognitive systems function well for cancer survivors. By contrast, under conditions of increased demand, cognitive failures may be more frequent and produce negative consequences. It is at this point where the survivor experiences a perception of disparity between memory demands in the environment (such as work demands) and available resources (lowered perceived cognitive abilities) that leads to increased arousal and distress [65]. This arousal and distress can lead to reduced cognitive performance through autonomic interference with cognitive systems of attention or recall. 

MAAT directly addresses this cycle triggered by the disparity between perceived demands and perceived cognitive resources (abilities). It does so by enhancing survivor awareness of this psychological process and by building cognitive coping capacities. MAAT has 4 components to achieve this: (1) education [66,67]; (2) self-awareness (self-monitoring to help BCS identify “at risk” situations where memory problems are likely to arise); (3) compensatory strategies (such as verbal rehearsal, self-instruction, keeping an organized schedule); (4) stress management (applied relaxation training, cognitive restructuring). These components work in concert to produce improved neurocognitive and QOL outcomes [53,54,68]. Detailed discussion on the theoretical underpinnings of MAAT and the mechanisms of behavioral and cognitive change is in the introduction of the clinician manual [28].

### 2.2. Data Capture

With existing software, we have developed a data capture method to enable tracking of MAAT outcomes for cancer survivors anywhere in the world where there is internet access. Research Electronic Data Capture (REDCap, Version 18.8.3, 2023, Vanderbilt University; https://www.project-REDCap.org/ (accessed on 24 July 2023)) is a secure online service that can build and administer online surveys and store collected data. It is endorsed by many university-based or other research organizations [69]. It is also an appropriate adjunct to telehealth clinical service research [70,71] and can remove the data collection burden from the busy clinician and minimize the questionnaire burden on survivors. REDCap can automatically send survivors a secure link through their personal e-mail account, where they can access CF8 and/or other measures through a simple mouse click. We developed a REDCap program using CF8. We have provided a description of setting up a similar REDCap program or “project” in the Appendix A. 

Once the clinician and survivor agree on MAAT or other CRCI treatment as a plan of care, steps for enrolling individual survivors in outcomes monitoring involve:(1)The survivor provides his/her individual e-mail address (an unshared, private account);(2)The address is entered into a secure link to the REDCap outcomes monitoring system;(3)REDCap then sends an e-mail to the survivor with a secured link to complete informed written consent (IRB-approved or institutionally approved as a clinical improvement project);(4)After signed consent is obtained, REDCap presents the survivor with a one-time demographics form assessing variables that could affect MAAT outcomes (e.g., age, type of cancer, year of diagnosis, type of treatment(s), and year of cancer treatment completion). These data can help identify which types of cancer survivors respond to MAAT and can be helpful for clinic management and target resources;(5)Survivors then complete PROMIS outcome measures via weekly e-mails. No identifying information, such as e-mail address or name, is linked to responses, demographics, or PROMIS measure data. REDCap automatically supplies a digital identifier. This identifier can be used by the clinician to identify and track individual patient outcomes.

Our REDCap/MAAT outcomes monitoring system uses US Health Insurance Portability and Accountability Act (HIPAA) “Safe Harbor” identification guidelines. Safe Harbor guidelines are based on the principle that no data are collected that would allow an individual to be identified if stolen data were cross-referenced with publicly accessible data (https://www.hhs.gov/hipaa/for-professionals/privacy/special-topics/de-identification/index.html (accessed on 2 March 2023)) [72]. For example, no addresses, dates of birth, units of time smaller than one year, etc., are asked for so that these cannot be cross-referenced with potentially publicly available information such as a birth certificate or other documents of a particular municipality (e.g., taxation or residential rolls). Once anonymous data are collected, REDCap can then export the data file (e.g., Microsoft Excel) to a secure server for analyses. Finally, an important point must be made about acquiring PRO data from patients for purposes of clinical service evaluation and clinical research. Emphatically, no survivor in clinical rehabilitative care is or should be, mandated to follow a SCED outcomes monitoring protocol. As with any consent process in healthcare or healthcare research, patients should be allowed to decline data collection procedures at any time or start MAAT at a time of their choosing in collaboration with their healthcare professional. We believe we have identified a streamlined consent process with our outcomes monitoring system that strikes a three-way balance between (1) the rights and dignity of survivors, (2) the autonomy and judgment of clinical professionals, and (3) the added value of outcomes data collection and analysis of CRCI treatment in clinical practice [73]. Contemporary guidelines for discussing the consenting process for outcomes data collection in a “learning healthcare system” are detailed in comprehensive reviews by Clapp and Fleisher and Kass and Faden [74,75].

### 2.3. Analyzing, Interpreting, and Reporting Results

In SCED, there are two methods of analyzing and interpreting data collected: (1) visual inspection and (2) statistical analysis [42]. Methods of visual inspection include:(1)Level, or simple quantity of, data in each phase on the vertical axis;(2)Trend, or slope, in each phase;(3)Variability within phases;(4)Overlap or how similar or different scores are between baseline and treatment phases;(5)Immediacy of Effect, or how rapidly a survivor may improve cognitive function in phase B.

There are a number of SCED statistical methods used to determine significant differences between baseline (A) and treatment (B) phases [42]. The Tau-U statistic is a test of “non-overlap” between baseline and treatment phases [42,44,45]. Baseline-corrected Tau (BC-Tau) affords more robustness using Kendall’s Tau rank correlation coefficient to evaluate A-B non-overlap [43]. A BC-Tau calculator specifically designed for A-B designs is available at www.ktarlow.com/stats/tau (accessed on 25 April 2019) [76]. BC-Tau is expressed as a coefficient bound by −1 to +1 and can be interpreted as an effect size, and its use in SCED adds confidence to visually observed A-B differences between baseline and treatment phases [42,43,45]. A value greater than zero indicates a positive association between the treatment phase and the outcome variable, corrected for any baseline trend, if necessary, as illustrated in the case example below.

A case Illustration from the primary author’s (RJF) clinical practice Is detailed here using SCED reporting guidelines. This includes a brief description of survivor characteristics (e.g., cancer diagnosis, cancer treatments, medical comorbidities that can affect cognition), how and when the intervention (MAAT) was administered, and the outcome measures used [40,51].

## 3. Results

### 3.1. Case Example

Figure 4 presents the case of a 44-year-old, premenopausal female breast cancer survivor initially diagnosed with ductal carcinoma in situ (DCIS) of the left breast. She underwent a prophylactic double mastectomy with reconstruction. Genetic profiling indicated she was BRCA 1 and 2-negative, and surgical outcome indicated no evidence of disease with complete remission. However, 4 years later, she was found to have ER+/HER2+ disease in the left breast. She underwent neoadjuvant chemotherapy with docetaxel, carboplatin, trastuzumab, and pertuzumab, followed by surgery and radiotherapy, prior to daily Tamoxifen (20 mg/day) with Lupron intramuscular injection every three months. These chemotherapy agents and hormonal therapies were identified as adversely influencing cognition [1,15], with combined chemotherapy/endocrine therapy showing greater effects [77]. Neuropsychological testing was completed by an independent clinical neuropsychologist prior to referral to our clinical service. Her intellectual functioning was high, with a full-scale IQ of 117 and her general cognitive abilities tested at 121 (standard score). She showed mild to more severe impairments in processing speed relative to her strong general neuropsychological abilities. For example, a standard score on letter cancellation was 90, with standard scores of 73 on Stroop color naming and 87 on the Stroop color–word tests, respectively. In our clinical evaluation, she scored in the normal range for anxiety (GAD-7 = 5) and depression (PHQ9 = 4), suggesting no anxiety or mood disorder.

Processing speed problems were exacerbated by anxious responses under time and performance demands. As a library teacher of elementary school students, the survivor reported complaints of being unable to keep up with conversations, respond efficiently to student behavior, and maintain focus on classroom instruction. This was especially worse in the latter half of the school day when she was more cognitively fatigued. She instructed different classes of children of varying ages, requiring modification of the style of classroom instruction and behavioral management to suit student age levels. Her report of struggles of “keeping up” under these varying demands was consistent with the MAAT diathesis–stress conceptualization of the functional impact of cognitive impairments detailed above. These problems led to a reduction to a half-time employment position. 

We agreed upon a treatment plan with MAAT, and she completed CF8 weekly beginning the day of our initial evaluation. She also completed a modified FACT-Cog impact on quality of life (IQOL) scale, adopting the same Likert scale as CF8 (higher scores = better QOL; 5 = never; 1 = very often (several times a day), raw score range of 4–20). We used raw scores on this scale, as there is no current T-score conversion. We also had a goal of gradually resuming a full-time teaching schedule. It is important to note that while she was on hormonal therapy and Lupron injection, Lupron was constant throughout the treatment plan. She did switch from Tamoxifen to Arimidex after the 10th data point (2 July; Figure 4) and after her best CF8 score. This change may have had a limited impact on CF8 outcomes since most gains occurred while she was taking Tamoxifen. However, unknown endocrine level variation could occur with uncertain effects on CF8 scores. Due to her travel and work schedule, she completed MAAT visits every other week, and we condensed MAAT content from eight to five visits to minimize the time and travel burden. Prior to the COVID-19 pandemic, her health insurance plan did not reimburse “video visits”, which our service did, in fact, have available at the time. Telehealth is presently utilized by a majority of patients in our behavioral oncology service at UPMC Hillman Cancer Center. While her case may be common in clinical survivorship service, she would have been excluded from enrollment in MAAT trials given recurrent cancer, and she would have been dropped from trial participation given the modification to the MAAT schedule. Therefore, the A-B SCED outcomes presented are representative of real-world MAAT outcomes.

### 3.2. Visual Inspection and Statistical Analysis of Outcome Data

Visual inspection of the level, trend, variability, non-overlap between phases, and immediacy of effect was the first step of the analysis. In Figure 4, the survivor appears to have improved cognitive symptoms and IQOL in the treatment phase (B) over baseline levels (A). The trend appears to move upward as MAAT begins, and there is some, though not excessive, variation within each phase (variability). The baseline data do not appear to overlap with treatment phase data, and there is some evidence of immediacy of benefit seen over the first three visits on both the CF8 and IQOL scales. Note, too, that the change in CF8 twice exceeded the MCID of five T-score points from baseline (25.73) to the last data point at follow-up (37.93), indicating clinically significant and meaningful change. 

For statistical analysis of cognitive function (CF8), over the phase A trend (baseline), BC-Tau = 0 was not significant (*p* = 1.0), as there was no variability in the three scores over three weeks. With no correction needed for baseline trend, the baseline vs. treatment phase comparison for CF8 scores was completed. There was a significant A-B difference (Tau = 0.764, *p* = 0.017; *SE*_Tau_ = 0.289), indicating a statistically significant improvement in perceived cognitive function with MAAT. The magnitude of Tau demonstrated a large treatment effect (0.764). For IQOL, the phase A baseline trend was not significant (BC-Tau = 0.816, *p* = 0.540). Thus, an uncorrected baseline vs. treatment phase difference was calculated (BC-Tau) and was significant (Tau = 0.743, *p* = 0.018; *SE*_Tau_ = 0.299), again indicating a significantly improved IQOL score with MAAT.

While the MCID was achieved in CF8 and cognitive symptoms and self-reported QOL improved, the last T-score data point on the CF8 score (37.93) was 1 SD below the population mean of 50, indicating persistence of symptoms. Finally, with respect to the goal of returning to full-time employment, she remained in part-time employment.

## 4. Discussion

This article outlines an outcomes monitoring system using REDCap data capture software, cost-free measures from PROMIS, and SCED methods to evaluate survivor CRCI treatment outcomes. In the breast cancer survivor case example, the MCID was achieved in self-reported cognitive function and IQOL over the course of MAAT and statistical significance was also observed. MAAT appeared to remain effective for this survivor for CF8 outcomes even when presented over fewer sessions to accommodate her schedule as she began to resume social roles post-cancer treatment. 

The outcomes monitoring system and SCED methodology detailed here could be used with multiple cases for purposes of replication and data aggregation across numerous cases. This would help answer questions about MAAT effectiveness or other CRCI treatments in clinical practice and help cancer rehabilitation facilities evaluate their programs. Moreover, the data capture and analysis methods described here could help multiple services aggregate case data and assess which types of survivors with various types of cancer and/or treatment (e.g., chemotherapies, oral agents, stem cell transplant) respond favorably to CRCI treatment and which types of survivors may not. In short, cognitive symptom outcomes monitoring can provide a system by which clinical managers can more objectively evaluate treatment services to aid resource allocation [78]. For example, it may be that while chemotherapy recipients tend to respond favorably to CRCI treatment such as MAAT, individuals who undergo intracranial irradiation may respond more favorably to pharmacotherapy or computerized interventions. SCED data aggregated across cases could provide an improved understanding of effective clinical options for the management of CRCI and inform oncology practitioners on how to best approach the problem in day-to-day practice [23,24].

In the era of precision medicine, SCED methods are being examined as possible ways to evaluate genomically tailored cancer treatments in clinical oncology settings [79,80,81,82]. Some newly developed immune or targeted therapies may be administered to such low numbers of individuals with certain genotypes that there is not adequate statistical power to evaluate the effects of CRCI treatment with a standard RCT [22]. Using a cognitive outcome monitoring system such as that outlined here could help evaluate CRCI treatment effectiveness in the relatively few individuals with CRCI who have either been treated for rare cancers or who have undergone novel cancer therapy.

There are limitations to the cognitive outcomes monitoring system presented in this article. First, while REDCap (and other data capture software) has the capability to automatically remind survivors to complete measures, it does not automatically organize data or transfer it to highly usable databases for statistical analyses without editing or managing data, such as providing logical data labels to variable columns or fields. Identifying trained staff to manage data is critical, but busy clinical services may not have available personnel. This is one of several barriers to PRO use in cancer clinical practice [30]. As data capture software improves, better automation of data management systems will likely evolve. Second, survivors who do not have internet access due to cost or who live in remote regions where it is unreliable may not be able to benefit from the convenience of remote outcomes monitoring or CRCI treatment. There is a growing use of local pharmacies as networks to enhance internet access for telehealth, especially for those who live in rural regions or who have only occasional access to the internet with smartphones [83,84]. These and other access methods will require continued development to help close digital access disparities. Finally, our proposal to use SCEDs to provide information on CRCI treatment effectiveness does not provide a comprehensive interpretation of individual data, or that aggregated across numerous cases. SCEDs and PROs can provide valuable information about CRCI treatment effectiveness, but accurate interpretation of outcomes can provide a fuller picture of treatment strengths and limitations. For instance, while the case example presented illustrates some meaningful clinical improvement in cognition and QOL impact after MAAT, her occupational outcomes (not assessed by PRO in this case) were not improved. Careful review of PROs and the limits of what they can measure is important in accurately understanding CRCI treatment impact in clinical settings [30]. However, overall, the outcomes monitoring system proposed here can help contribute to building a knowledge base of the state of clinical offerings for CRCI management as they continue to be developed, evaluated, and disseminated. Such information can add to the evidence base to help cancer care clinicians and patients make informed care choices [23,24].

## 5. Conclusions

In closing, we have described a system of continuous evaluation once an evidence-based CRCI treatment, such as MAAT, is deployed in cancer rehabilitation settings. Given the growing cancer survivor population globally, and thus the growing impact of CRCI on survivorship disease burden [10,27,85], we urge clinicians and researchers to continue to develop and refine this system to efficiently evaluate real-world CRCI clinical outcomes. It is the intent of this article to encourage clinical professionals to consider using elements of this system in their own clinical practice for CRCI treatment. This will ultimately help inform the options available for CRCI management in cancer care.

## Figures and Tables

**Figure 1 cancers-15-04643-f001:**
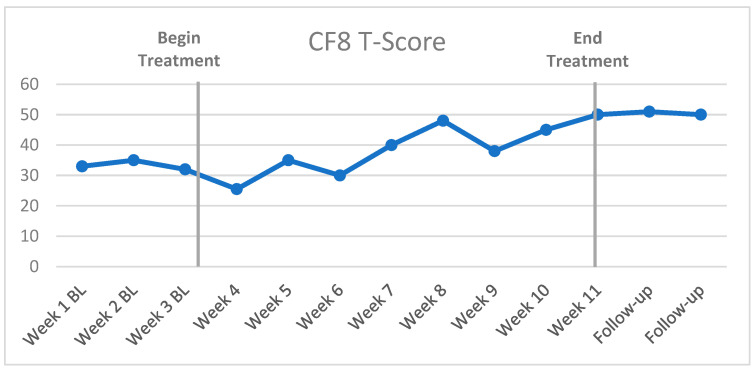
Example of CF8 outcomes in an A-B SCED for CRCI treatment.

**Figure 2 cancers-15-04643-f002:**
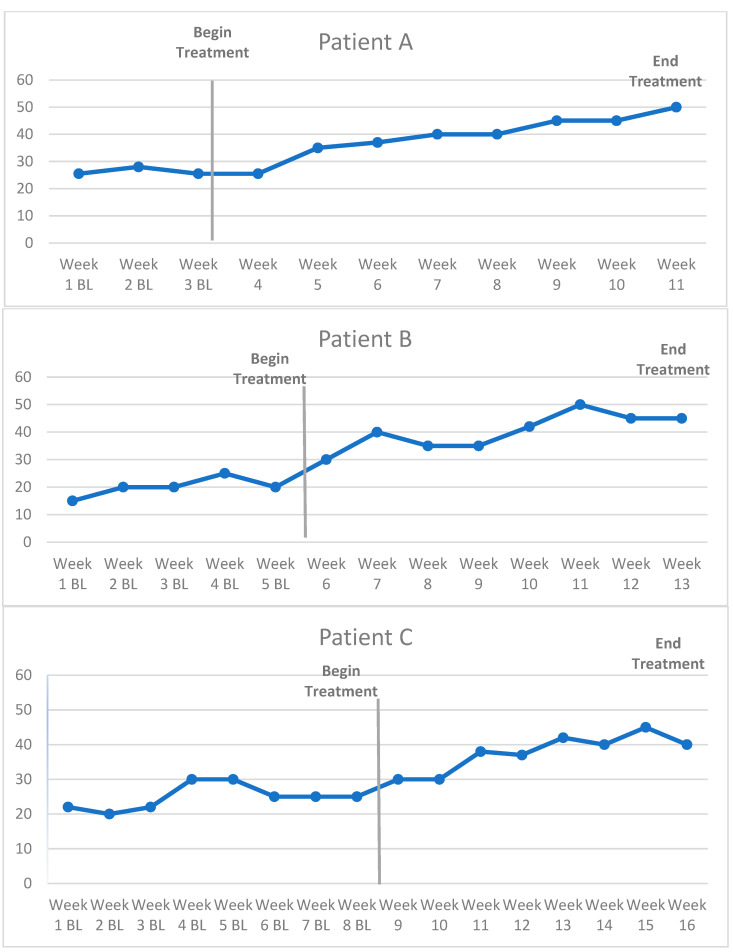
Example of CF8 outcomes in a multiple-baseline across-subject design.

**Figure 3 cancers-15-04643-f003:**
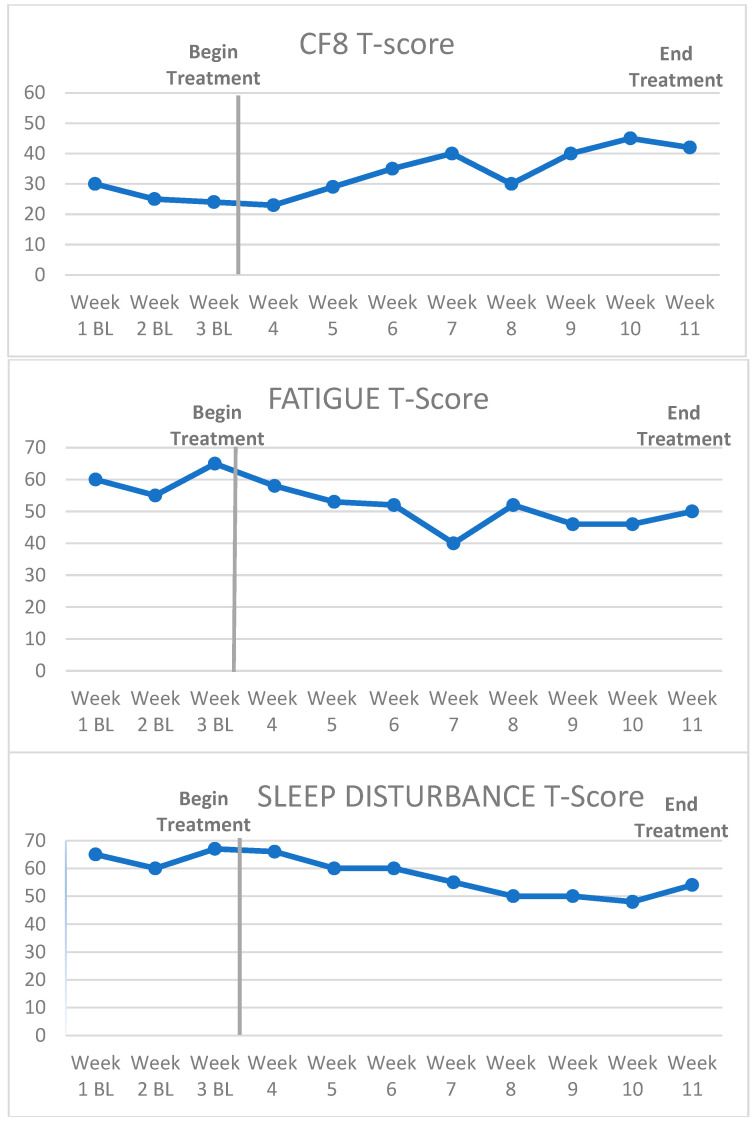
Example of a multiple baseline across symptoms design.

**Figure 4 cancers-15-04643-f004:**
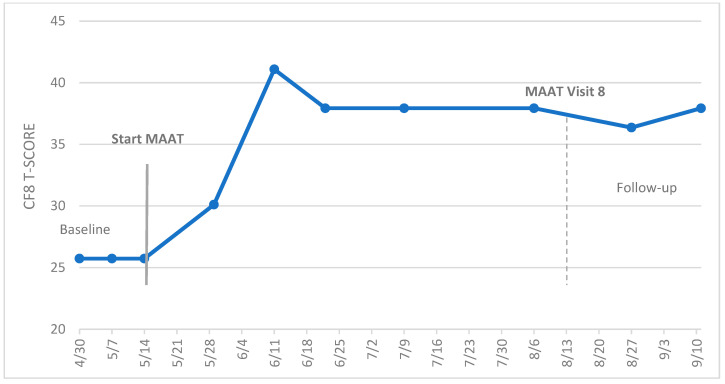
Example of MAAT cognitive function (CF8 T-score) and impact on quality of life (IQOL) outcomes in an A-B SCED.

## Data Availability

The data used in the case example are not publicly available due to patient privacy protections. However, raw de-identified data by MAAT visit (dates of visits are excluded) are available upon written request to the corresponding author.

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
