# Peer review of "Using Single-Case Experimental Design and Patient-Reported Outcome Measures to Evaluate the Treatment of Cancer-Related Cognitive Impairment in Clinical Practice"

_cancers, 2023, doi:10.3390/cancers15184643_

Round 1

Reviewer 1 Report

This is a strongly-written manuscript which presents a methodology including detailed, replicable instructions alongside an applied example in order to maximize impact. The authors make an effective argument for the benefits of single-case experimental design in order to advance our understanding of treatment effectiveness in real-world clinical settings outside of the eligibility constraints and financial investment required to carry out an efficacy randomized controlled trial. This is consistent with the NIH ORBIT Model of intervention development which promotes use single-case experimental designs in order to compile preliminary evidence for the effectiveness of an intervention. I have no substantial revision requests and would recommend this manuscript for publication.

Thank you for the opportunity to review this paper. 

Reviewer 2 Report

This paper examines an interesting topic, but there are some points that need revision: The number of diagrams is not helping the reader understand the findings. Please explain why the single case experimental design is of importance. Methodologically, a full scale study is more useful.

Cognition can be influenced by hormonal changes as well (see two case studies: Giannouli, V., Toulis, K. A., & Syrmos, N. (2014). Cognitive function in Hashimoto’s thyroiditis under levothyroxine treatment. Hormones, 13, 430-433.). Please discuss this point and add the testing results for the included patients regarding hormonal status.

How can you claim that MAAT (which is not describing in enough detail to the reader) is the only factor influencing the variables? Are there any other third factors (apart from hormones) intervening?

It is not clear to the reader why these outcome variables were chosen instead of other tests. Please justify.

The discussion should be more detailed and supported by relevant literature.

T or Z scores conversion, please discuss.

Minor editing needed.

Reviewer 3 Report

Using Single Case Experimental Design to Evaluate Treatment of Cancer-Related Cognitive Impairment in Clinical Practice

This study provides an illustration (with open resources) of using a SCED approach to assessing a CRCI intervention. Overall, the article is noteworthy and the approach is solid. However, the article can be improved in a variety of ways. I have provided specific areas and suggestions for improvement below:

Title: I suggest adding “: An illustrative Example” after the current title.

Introduction:

·        The authors appropriately list the multiple potential causes approaches of CRCI as well as the multiple interventional approaches, including CBT. They then delve into their CBT intervention of interest (MAAT). However, between listing the mechanism and different interventional approaches there needs to be more providing a rationale for honing in on MAAT. First, why is CBT a useful approach to focus on compared to the other approaches? Second, why is MAAT an important approach of CBT to focus on? lastly, what are the theoretical mechanisms of action of CBT (and specifically MAAT) on CRCI improvement (e.g., compensating for lower cognitive capacity, processing speed, etc through practical mechanisms. You touch on this, but more is needed especially in cognitive theory)?

·        Please provide more background on SCEDs. This is the focus of this work. Provide examples of other work (doesn’t need to be in cancer, for instance, psychiatry) using this approach and what sort of outcomes can be had as well as their practical importance. You have one sentence of this at the end of the introduction, however I suggest moving this further up, (after first mention of SCEDs) and providing more detailed examples, and rather ending with a strong rationale statement or two for your research. For instance, build upon your first sentence in the last paragraph of your introduction.

·        Overall, this is a very good idea and the individuality of cognition in itself is an important area of consideration (See Haywood et al., 2021). However, I feel as though your currently presented introduction does not do justice to your rationale. I also feel both further interdisciplinary and oncology literature is required to fully justify the utility of the research. Recommended references below. But there are additional relevant works.

·        I suggest ending with aims.

Unmet supportive needs (including cognition): Fan, R., et al., Unmet supportive care needs of breast cancer survivors: a systematic scoping review. BMC Cancer, 2023. 23(1): p. 1-24.

Methods:

·        The outcome measures selected are appropriate but the section is very hard to follow. I suggest minimising non required text here and also providing some indication of the reliability and validity of the measures.

·        Overall the methods section is thorough but sometimes hard to follow. I suggest adding a figure regarding the assessment timepoints (including multiple baselines. As random give an example).

Results:

·        Results section is very well written and the figures are easy to understand. An important concern I have is for the reader to see the potential of ‘cherry picking’ a consumer that has shown positive results from the intervention for this example. I understand that showing the efficacy of the intervention is not the primary point of this paper. As I suggested above, a clear statement of the research aim in the introduction would minimise this. However, I also suggest a statement or two within this results section emphasizing that this is a worked example of the SCED method used to explore the efficacy of MAAT on the individual level, and not an assessment of the efficacy of MAAT generally (as you say there are RCTs coming for that)

Discussion:

·        Again within this section, it is necessary to ensure that the reader does not think that this is a primary text of the efficacy (due to the potential to pick a particular consumer to show positive effects). I suggest adding in “for this consumer” before “MAAT appeared to remain effective for this survivor even when presented over fewer sessions to accommodate her schedule as she began to resume social roles post-cancer treatment”

·        This section requires additional discussion around the use of SCEDs in other disciplines and the similarities and potential differences to this context. What are some considerations when using this approach, for this purpose, within the clinical setting?

·         You speak how this might be useful to the clinical setting by aggregating such data. However, you have previously emphasised the utility of this also on the individual level. How might this data be used in the clinical setting on the individual level to inform clinical decision making?

·        There has been recent work regarding health professionals’ perceptions, experiences and challenges relating to service provision regarding CRCI, I suggest making reference to this and how this particular approach may lessen these service provision difficulties and how these difficulties may impact the interpretation and use of this data (positively and negatively). I have provided two references below that are particularly useful to this point. 

 He, S., et al., Australian oncology health professionals’ knowledge, perceptions, and clinical practice related to cancer-related cognitive impairment and utility of a factsheet. Supportive Care in Cancer, 2022. 30(6): p. 4729-4738.

Haywood, D., et al., Oncology healthcare professionals' perceptions and experiences of 'chemobrain' in cancer survivors and persons undergoing cancer treatment. General hospital psychiatry, 2023: p. S0163-8343.

Overall

·        Overall, this is an interesting article that provides an approach that facilitates many important aspects of contemporary survivorship care (e.g., individual-level care and support). The provision of open resources is also excellent. The introduction could be strengthened to give better credit to the rationale. The results need to emphasise that this is an example, and not a primary assessment of MAAT efficacy. The discussion needs to place this approach within the literature better. Including within the current understandings, perceptions, experiences, and challenges f health professionals (who would be interpreting the data).

Round 2

Reviewer 3 Report

I would like to thank the authors for their constructive perspectives taken from my comments. They have adequately addressed my concerns and suggestions. I recommend acceptance in its current form.